# Consumers discard a lot more food than widely believed: Estimates of global food waste using an energy gap approach and affluence elasticity of food waste

**Monika van den Bos Verma** [ORCID] *, **Linda de Vreede**[¤a], **Thom Achterbosch, Martine M. Rutten**[¤b]

Wageningen Economic Research, Wageningen University and Research, The Hague, The Netherlands

¤a Current address: Erasmus University Rotterdam, Rotterdam, The Netherlands
¤b Current address: Dutch Ministry of Foreign Affairs, The Hague, The Netherlands
* monika05@gmail.com

## Abstract

This work provides an internationally comparable consumer food waste dataset based on food availability, energy gap and consumer affluence. Such data can be used for constructing meaningful and internationally comparable metrics on food waste, such as those for Sustainable Development Goal 12. The data suggests that consumer food waste follows a linear-log relationship with consumer affluence and starts to emerge when consumers reach a threshold of approximately $6.70/day/capita level of expenditure. These findings also imply that most empirical models overestimate consumption by not accounting for the possibility of food waste in their analysis. The results also show that the most widely cited global estimate of food waste is underestimated by a factor greater than 2 (214 Kcal/day/capita versus 527 Kcal/day/capita). Comparison with estimates of US consumer food waste based on national survey data shows this approach can reasonably reproduce the results without needing extensive data from national surveys.

## Introduction

It is a widely held and cited belief that one third of all food available for human consumption is lost or wasted [1]. There is a clear distinction between food loss and food waste [1]; with part of the latter attributable to consumers. One problem with the estimates of food waste is that they consider only part of the picture. If we look at food waste (FW) as an outcome of the food system, then there are supply side determinants but there is also a demand side story [2]. We cannot waste more than the food available to eat, therefore consumer food waste is constrained by an upper limit as dictated by food availability (FA) as determined by supply side factors. Apart from being questionable for reasons identified in [3–6], by assuming that a fixed factor of available food is wasted [7], the supply side is the only part of story captured by the Food and Agriculture Organization (FAO) estimates. In the FAO methodology, consumers play no

**Data Availability Statement:** All relevant data are within the manuscript and its Supporting Information files.

**Funding:** MV, TA and MR received funding from Dutch Ministry of Economic Affairs (project grant# KB-22-002-005) and grant number KB-1-1C-1. The funders had no role in study design, data collection and analysis, decision to publish, or preparation of the manuscript.

**Competing interests:** The authors have declared that no competing interests exist.

part in determining estimates of consumer food waste. To account for the demand side, one needs to look at uses of food available for human consumption (waste being one), and factors that determine how much food is wasted by consumers. The demand side requires data on consumers' socio-economic attributes such as income, education, residence, food-culture etc. There are individual attempts to capture the impact of these consumer specific attributes [8,9] using regression methods, but no studies have attempted this at a global level. While we found work using global age and sex distribution to impute global FW [10], a measure of responsiveness of FW to these demographic factors as provided by regression methods, was found missing. With the aim to fill such gaps in research, this work is a first attempt at linking the amount of food wasted to one such consumer attribute. The choice of attributes can be wide but as a first step we start with the most basic of all—consumer affluence. Consumer affluence is an often cited but never quantified determinant of food waste; with claims that the richer populace waste more than their poorer counterparts [1,11].

In the process of capturing the demand side of the food waste story we gain several other helpful insights. We find that:

1. FAO figures grossly underestimate the extent of food wasted by consumers;

2. The relationship between consumer affluence (as measured by consumer expenditure) and food waste can be approximated by a linear-log functional form which helps us identify a threshold level of consumer affluence beyond which food waste rises rapidly;

3. This identified relationship can be used to generate a globally consistent dataset requiring limited and readily available macro-economic information. The dataset can be used to construct globally comparable indicators and metrics for Sustainable Development Goals (SDG12);

4. The analysis introduces a new concept—the affluence elasticity of food waste and shows that it increases rapidly at first but then tapers off as affluence increases;

5. The new elasticity concept and associated findings have implications for the theory of income elasticity of consumption, and insights for policy practitioners.

## Materials and methods

As FAO food waste estimates cover only part of the story, it renders the estimates unusable for identifying the relationship between consumer FW and consumer affluence. We therefore impute FW data in the following manner (Part 1) and then use the data thus obtained to establish a relationship between FW and consumer affluence (Part 2). The two subsections are followed by description of data required to implement the method.

### Part1: Calculating sample FW data

Conceptually, we start by looking at FW as a result of their decisions after food is made available to consumers. Once the food is made available to consumers, it can either be consumed or not eaten. For the purpose of this study we classify any food fit for human consumption, but not eaten, as waste. In other words, second best uses are inefficient uses of resources and therefore considered waste. Fig 1 provides a simplistic representation of the approach.

The actual method used to calculate FW is adapted from [3] and follows the energy gap approach [12]. Following the human metabolism model, energy from food consumed is spent on maintaining the current body weight (BW) and Physical Activity Levels (PAL) associated with one's lifestyle. Any excess energy intake manifests itself in increased BW overtime (Fig 1).

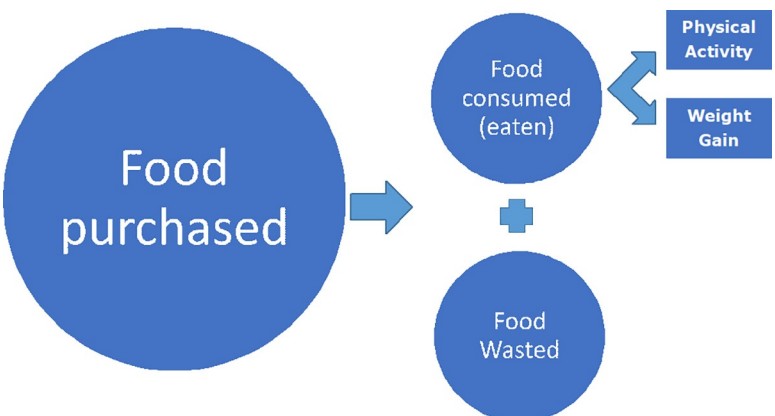

**Fig 1. Relationship between food availability and food use (consumption/eat and waste).**

Given that we focus on a given point in time, rather than over time, we calculate the energy requirements (ER) of populace needed for maintaining their current BW and PAL. The difference between the ER thus calculated and the FAO data on FA for human consumption is then used as an estimate of FW, at a given point in time, using Eq (1).

$$FW_c = FA_c - \boxed{PAL_c * \left[ \sum_{a,g}(\alpha_{a,g} * ps_{a,g,c})BW_c + \sum_{a,g}(c_{a,g} * ps_{a,g,c}) \right]} \qquad (1)$$

The boxed term in Eq (1) is the expression for ER obtained through food consumption. Subscript 'c' denotes a sample country, '$\alpha_{a,g}$' and '$c_{a,g}$' are age and gender specific coefficients for calculating ER of a give sample population. '$ps_{a,g,c}$' are population shares by age and gender in a given country. FA, ER and therefore FW are measured in kilocalories per capita per day (Kcal/cap/day). Eq (1) is used to obtain FW estimates for an average consumer in all countries in our sample. Further details on derivation and application of the Eq (1) are provided as supporting information (SI) in S2 File.

## Part2: Quantifying the relationship between FW and consumer affluence

As the calculated FW data pertains to a limited sample, it cannot be used to infer global FW. However we postulate that using this sample data allows a relationship to be identified between consumer affluence and their FW. Essentially, we argue that if:

1. Food availability (FA) is a function f(.) of consumer affluence (Y);

2. Food Consumption (FC) is a function g(.) of consumer affluence (Y);

3. FA = FC + FW.

Then using the three assertions above, one can argue that Eq (2) follows and FW in a country is a function h(.) of the level of its consumers' affluence.

$$FW_c = FA_c - FC_c = f(Y_c) - g(Y_c) = h(Y_c) \qquad (2)$$

This relationship between FW and consumer affluence can be used to obtain an estimate of global FW, and estimates for out of sample countries and years. An implicit assumption underlying the out of sample and future predictions of FW is that the evolution of population BW and FA follows same trends in poor nations as in rich ones. Given the genetic differences [13], land pressure, climate change induced drop and variability in agricultural yields etc., this

assumption is not perfect, but necessary in absence of any further country specific information on such trends. Furthermore, the analysis focuses on averages while evolution of FW also depends on distribution of affluence within countries' populations. The higher the degree of inequality, the more the estimates are likely to diverge from ground reality.

**Implications for theory of consumption elasticity.** In addition to providing consumer FW estimates that evolve with affluence, this approach enables the introduction of a new concept–affluence elasticity of waste–as a measure of responsiveness of consumer food waste to consumer affluence. Our objectives in introducing the concept are to draw attention to the importance of thus far ignored affluence elasticity, highlight its implications, and argue for its importance in the food waste and consumption literature.

The approach essentially distinguishes between calories purchased (through FA), calories consumed (through ER), and calories wasted (FW) by consumers. Taking an example of a single consumption commodity i, let the associated calories purchased, consumed, and wasted be $CP_i$, $CC_i$ and $CW_i$ respectively, such that:

$$CP_i = CC_i + CW_i$$

Its equivalent in physical commodity units can be written as

$$a_i.QP_i = a_i.QC_i + a_i.QW_i$$

where $a_i$ is the calorie content per physical unit of commodity i and $QP_i$, $QC_i$, $QW_i$ are the purchased, consumed and wasted quantities respectively. Differentiating with respect to expenditure (E)–a measure of affluence—yields:

$$\frac{\partial QP_i}{\partial E}\frac{E}{QP_i} = \left[\frac{\partial QC_i}{\partial E}\frac{E}{QC_i}\right]\left(\frac{QC_i}{QP_i}\right) + \left[\frac{\partial QW_i}{\partial E}\frac{E}{QW_i}\right]\left(\frac{QW_i}{QP_i}\right)$$

Denoting elasticities by letter η, the above can be rewritten as an expression showing total expenditure or purchase elasticity ($\eta_p$) as a sum of consumption($\eta_c$) and waste($\eta_w$) elasticities where θ is the share of the purchased commodity that is actually consumed (commodity subscript i is dropped for simplicity of illustration):

$$\eta_p = \theta\eta_c + (1 - \theta)\eta_w \tag{3}$$

Standard consumption theory (not accounting for waste) mistakenly treats the purchase elasticity($\eta_p$) as elasticity of consumption($\eta_C$). While there does not seem to be major consequences of this mistaken identity for very poor nations, for the rapidly developing countries it leads to an upward bias in the consumption response. Models mistakenly using purchase elasticity as consumption elasticity over-predict the consumption response as they do not account that some of the food purchased is actually wasted and not eaten. Computable general equilibrium models know this problem well and devise ways to navigate it [14]. A decomposition similar to Eq (3), for elasticity of calories can be undertaken to decompose it into elasticity of calorie consumption and affluence elasticity of calorie waste.

## Data requirements

Details on the data used to implement this strategy for obtaining sample and global estimates of FW are described below.

**Food availability.** **FA** data for sample countries is retrieved from the FAO FBS (Food Balance Sheets) [15]. Food availability data (in Kcal/day/cap) is averaged for each country over the period 2001 to 2005 to rule out fluctuations in food production resulting from bad weather in any single year. We understand that FBS data is not free of problems, and particularly so for

lower income countries where subsistence farming is still prevalent [3–6]. Correcting for both supply and demand side problems in measuring FW is however, beyond the scope here. At present, only a single step towards including the demand side is undertaken in this work.

**Energy requirement.** Calculating the energy requirements needs data on several different components of Eq (1).There are three different **PAL values** used for countries, these three are associated with different lifestyle categories [16]: sedentary or light activity lifestyle (1.4–1.69), active or moderately active lifestyle (1.7–1.99) and vigorous or vigorously active lifestyle (2–2.4). These PAL values indicate the level of physical activity most often exerted by the population; values of 2.4 or higher are impossible to sustain over longer period of time and are associated with the lifestyle of professional athletes [16]. Developing and developed countries are believed to have very different lifestyles as it is generally believed that the population of developing countries has a higher activity level and therefore higher energy expenditures [17]. To avoid a subjective PAL assignment for countries, we use a PAL categorization based on the Human Development Index (HDI) which shows PAL and HDI to be negatively correlated [18,19]. The information is used to assign one of the three different PAL levels to countries based on their HDI scores: 1.4, 1.7 and 2 PAL values are assigned to sample countries with high, medium and low HDI scores respectively. While this assignment is not perfect, it is an improvement upon the assignment of the same PAL value to all countries as implemented by [10].

The last terms in the square brackets [.] in Eq (1) is the Basal Metabolic Rate (BMR) in Kcal/cap/day which denotes the energy required to sustain observed BW. The BMR term accounts for age and gender specific differences in energy requirements by using the country generic, but **age and gender specific coefficients** ($\alpha_{a,g}$ and $c_{a,g}$) as reported in [16]. Coefficients are aggregated using country specific age and gender **population shares**($ps_{a,g}$) from the World Bank Indicators dataset [20]. As the age categories as reported by World Bank and FAO data do not completely match, an approximate match is used. Specifically, population information from World Bank age categories 15–19 years, 20–29 years, 30–59 years and over 60 years are combined with BMR coefficient information from FAO age categories 10–18 years, 18–30 years, 30–60 years and over 60 years respectively, for both males and females. The population shares in Eq (1) are calculated based on the population aged 15 and older. Children are left out from the analysis as the BW estimates by the World Health Survey (WHS) [21] are also based on an adult population sample (aged 18+). Taking children into account should result in lower average energy needs of a sample population [10], particularly for countries with a relatively young demographic composition, which are also often identified to be low-income countries [22]. According to the human metabolism model, this might mean a lower ER and a higher FW.

Data on **BW** are available from the WHS [21], conducted in 2002–2004 in 70 countries to provide information on health systems and the health of populations. The WHS is a household survey covering more than 300.000 individuals. It provides data on the mean body weight in kilograms in 2003 for 67 countries. The three countries that are in the WHS but do not include information on the mean BW of the population are Australia, Ireland and Myanmar. In addition to the above mentioned three, two countries are omitted from the sample: The Dominican Republic and Comoros. The mean BW value reported for the Dominican Republic is much higher (125 kg) than the values reported for all other countries and very likely a data entry error in WHS. Comoros is left out due to missing FA data from FAO FBS.

**Consumer affluence.** While we don't need data on consumer affluence to calculate sample FW data, it is needed to identify the relationship between FW and consumer affluence. For **consumer affluence**, we use Annual per capita Actual Individual Consumption (AIC) data

**Table 1. Descriptive statistics and results for sample.**

| Variable | Bodyweight (Kg, 2003) | Food availability (Kcal/day/cap, 2001–05 average) | Food waste (Kcal/cap/day, 2003) |
|---|---|---|---|
| Average | 59.6 | 2704 | 351 |
| Minimum | 50.6 | 1,868 | 32 |
| Maximum | 76 | 3,757 | 1607 |
| Standard Deviation | 7.12 | 525 | 475 |

Source: Calculations using sample data

from the International Comparison Program (ICP) [23]. ICP ensures that this measure is comparable across countries.

Note that given the availability of BW data for only year 2002–2004, we are restricted to using the other data from roughly the same time-period (AIC data from ICP 2005 and FA data from 2001–2005) to quantify the relationship between affluence and food waste. Data for two of our sample countries (Guatemala and United Arab Emirates) is not available in the ICP 2005 database, which reduces our sample size used in Part2 of the methodology, to 63 countries. The complete dataset and details are available as S1 File.

## Results

### Sample results

A summary of the input data (BW and FA) and the resulting calculated FW data (Eq 1) for the sample countries is provided in Table 1. The values reported in Table 1 are population weighted averages of the sample countries.

For this sample, population weighted average energy requirement (given BW and PAL values) is 2427 Kcal/day/cap. In combination with FA data, this implies that on average consumers in the sample countries in 2003, wasted approximately 351 Kcal/day/cap (13% of calories available to consumers). Some countries in the sample show negative FW values; these are food deficit poor countries that barely meet their nutritional needs. The 351 Kcal/day/cap average excludes these negative FW values in the sample (the results for all sample countries are available in S2 File). The negative FW values for developing countries in this study are likely due to two reasons: a downward bias in FAO food availability data in these countries due to unaccounted subsistence production [6], which results in an intake that is higher than the reported availability and a resulting in a negative FW value; and not accounting for children in sample population could also contribute overestimating ER and therefore creating a downward bias on FW. Ours however is not the only study to find negative food waste values, [10] also find negative values based on the human metabolic model despite accounting for the under 18 population. They interpret the negative FW values as food deficit.

The maximum FW value of 1607 Kcal/day/cap in our sample is for Belgium, and the smallest positive number for FW in 2003 in our sample is found for Philippines (32 Kcal/day/cap).

### Relationship between FW and consumer affluence

**A linear-log relationship.** Using the FW data thus obtained, and ICP data on AIC for the sample countries, a simple data visualization (Fig 2) suggests a linear-log relationship between consumer FW and expenditure.

Regression coefficients of this suggested relationship are provided in Table 2 (columns 3 and 4). Using alternative variables as a measure of affluence–ln(GDP/cap) and ln(AIC/cap) from ICP–gives similar results but AIC shows a slightly better fit (Table 2, column 5). This

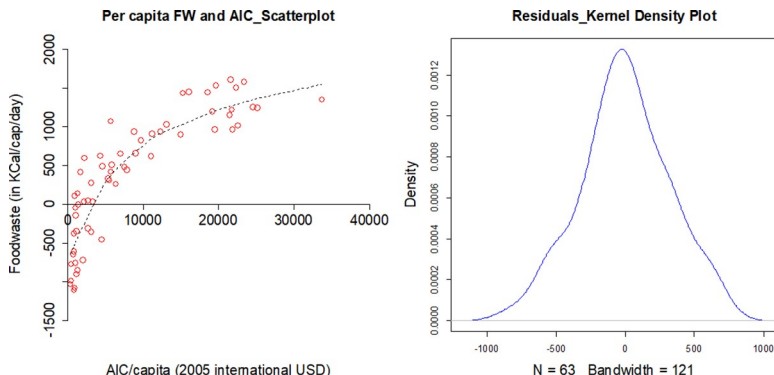

**Fig 2.** a) Left panel: Per capita sample Food Waste (FW) and annual per capita Actual Individual Consumption (AIC) b) Right panel: Kernel density plot of sample residuals from regression of per capita FW on natural log of AIC/capita.

shows that our coefficient estimates are robust to the choice of measure of consumer affluence. A kernel density plot of residuals with AIC/cap as measure of affluence shows them to be fairly normally distributed (Fig 2). Affluence is used as sole explanatory variable, as demand for calories/cap do not seem to be very responsive to food prices [24,25].

The negative estimates for the constant in the relationship show that FW is not a problem at low levels of affluence, in fact, using the AIC/cap estimates from Table 2, one can identify the income threshold (2450 International 2005 USD) beyond which consumer FW in a country turns positive and starts to increase rapidly. The positive slope estimates tell us that for a 1 percent increase in affluence, FW increases by about 5–6 (576.7/100) calories. Note that while changes in body weight over time don't explicitly feature in the regression equation and in obtaining the sample dataset, the sample variation in bodyweight across countries (Table 1) allows for this possibility.

**Estimates of FW and comparison with existing comparable literature.** As our sample represents only 67% of the world population, and some countries believed to waste a lot of food (including United States, Australia and Canada) are not present in our sample, we cannot use the population weighted average of sample FW data of 351 Kcal/cap/day (Table 1) as an estimate of FW for the world in 2003. We instead use the regression results (Table 2) and world AIC value (6095 international 2005 USD) from ICP 2005 to obtain an estimate of consumer FW for the world in 2005. This yields an estimate of 526 Kcal/day/capita for FW in 2005. At a global level FW rises to 727 Kcal/day/capita by 2011, accounting for 25% of calories available for human consumption. Note that these estimates of FW are derived using the central estimates of the regression coefficients in Table 2, and a range around these could be constructed using the coefficient ranges in the said table.

**Table 2. Coefficients of per capita food waste regression on affluence.**

| (1) | (2) | (3) | (4) | (5) |
|---|---|---|---|---|
| | | Constant | Slope | Model fit (R-squared) |
| **Alternative independent variable** | | | | |
| **ln(AIC/cap)** | Coefficient (95% confidence interval) | -4500 (-5054,-3946) | 576.7 (512,642) | 0.83 |
| | t-statistic | -16.23 | 17.74 | |
| **ln(GDP/cap)** | Coefficient (95% confidence interval) | -4537 (-5131,-3944) | 557 (490,624) | 0.81 |
| | t-statistic | -15.28 | 16.68 | |

Source: Own estimation using sample data

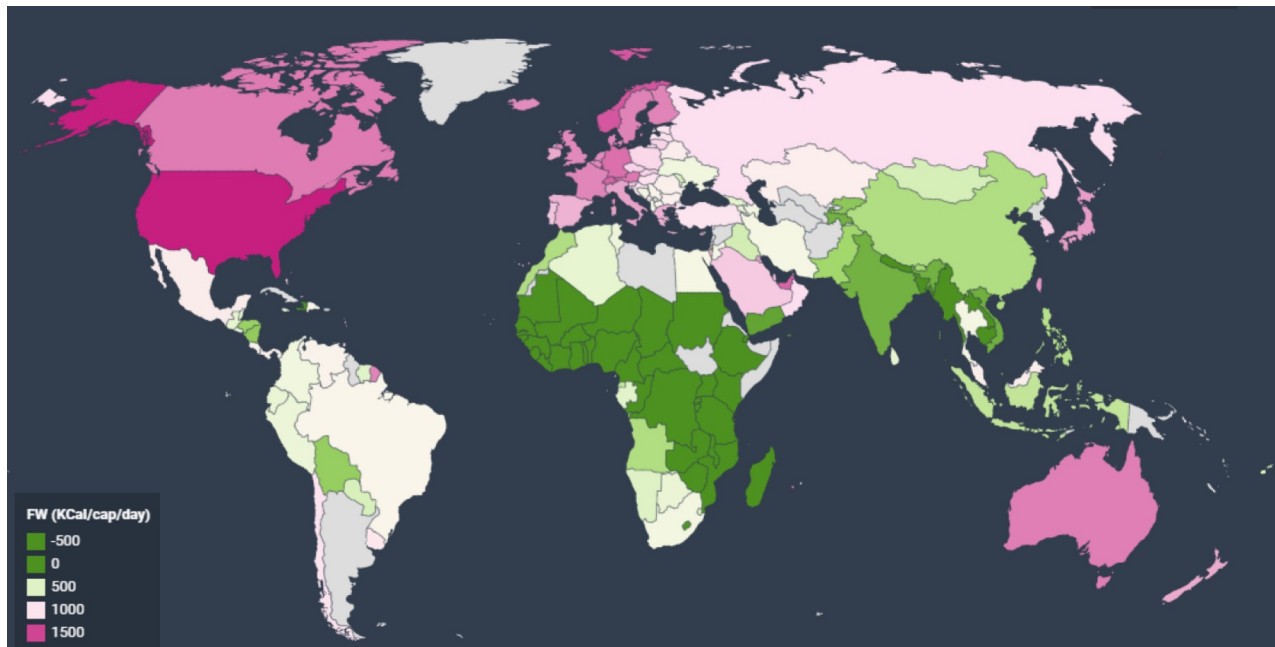

**Fig 3. Predicted food waste (Kcal/day/cap in 2011) for countries in International Comparison Program 2011 database.**

Using the same estimates (Table 2) and country specific AIC data for 2011 [26] we obtain FW estimates for all countries of world in 2011. The results are presented in Fig 3 below; in which the pattern across countries confirms the general belief that consumers in richer countries waste more food. The dataset underlying the figure is provided in SI (S3 File). This can be used as a consistent global consumer food waste dataset for developing metrics and indicators for inter-country/region comparison.

We also compare the current FW estimates with estimates of FW in the literature, with a particular focus on studies reporting Kcal estimates. While not exhaustive, Table 3 covers the relatively recent comparable work in the field. Kcal estimates [3,10,27,28] yield more readily to such a comparison. We do not directly draw a comparison with FAO physical waste estimates [1], as [27] translate those to provide Kcal estimates and can therefore, in essence, be seen as Kcal equivalents of the physical waste estimates. Due to different time and geographic

**Table 3. Comparison with kilocalories (Kcal) food waste estimates in comparable existing literature.**

| Existing comparable literature | Region/country of focus | Consumer FW estimate from literature: Kcal/day/cap (year) | Comparable affluence based estimates of FW from current work Kcal/day/cap (year) |
|---|---|---|---|
| Kummu et al. 2012[27] | World | 214 (2005–2007) | 526 (2005) |
| | World | 510 (2010) | 526 (2005) 727 (2011) |
| Hic et al. 2016 [10] | USA | 1050 (2010) | 1572 (2011) |
| | China | 620 (2010) | 329 (2011) |
| | India | 210 (2010) | 121 (2011) |
| Hall et al. 2009 [3] | USA | 1400 (2003) | 1482 (2005) |
| Buzby et al. 2014 [28] | USA | 1249 (2010) | 1572 (2011) |

Source: Compilation using estimates from the recent studies and current work

coverage, we use our regression coefficients (Table 2) to obtain our FW estimates for years beyond 2003.

**Comparison with Kummu et al.** [27]: If FAO's food availability estimates (2735 Kcal/day/cap) are correct, then globally consumers alone were wasting about 19% of calories available for human consumption in 2005. The corresponding number obtained by Kummu et al. using FAO food waste data, is in comparison only 8% (derived on the basis of numbers reported in Table 2 and Fig 2 in [27]).

Global FW estimates in terms of Kcal were first reported by Kummu et al. using data from FBS, and waste percentage assumptions [1]. Their analysis covers FW for cereals, fruits & vegetables, oilseeds & pulses, and roots & tubers, but not for all food (e.g. animal products are excluded). As per their study, over the period 2005–2007, of the total 614 Kcal/day/cap wasted in food supply chain, 214 Kcal/day/cap were wasted by consumers. The current study and [10] capture the total extent of FW in a manner which avoids errors in measurement and gathering of consumption data, and in conversion of physical weights to calories. Kummu et al.'s estimates are furthermore based on the waste percentages estimates [1] and are therefore subject to the same critique as the assumptions underlying these percentages. Also, Kummu et al. use food availability data which is already adjusted for country production losses, yet they account for production losses in estimating FW [10]. This, in addition to only partial coverage of food groups, results in a downward biased estimate of consumer FW.

**Comparison with Hic et al.** [10]: As with the current study, Hic et al. follow [3] to estimate food energy intake required to maintain the observed body weight. The current work however differs from theirs in certain aspects.

Hic et al. 2016 use FBS data on food availability to calculate a measure of FW but their energy requirement is imputed using body weight from different National Health Surveys (NHS) for 71 countries (filling the data gaps with weighted averages). They use three different PAL values to provide three alternative estimates of FW. While they provide estimates of the evolution of FW over 1965–2050, at global and country levels, they assume that BW over this period remain unchanged at the levels reported in NHS. Furthermore, unlike the current study, the BW data used from NHS for different countries comes for different time periods. For example, they use BW from 2011–2012 for Australia and from 1986–1992 for Canada, New Zealand, Sweden, Switzerland, Lithuania, Italy, Iceland, Israel and Serbia. Using a moderate PAL gives an average value of 510 Kcal/day/cap in 2011 for food surplus regions of the world and -120 Kcal/day/cap for parts of the world characterized by food deficit. 510 Kcal/day/cap is not very different from our 526 Kcal/day/cap (albeit for the year 2005), the current work, however, did not have to fill the gaps in data to get a global estimate. While Hic et al.'s global FW estimate of 510 Kcal/day/cap could be more precise than ours on account of including specific nutritional requirements of children, pregnant and lactating women, the claim cannot be made with certainty. This is because while taking pregnant and lactating women into account will increase energy requirements thus lowering FW levels, accounting for children (with lower calorie consumption than adults) will raise FW. The overall effect on relatively young countries is therefore ambiguous. At the same time it can be argued that the Hic et al. estimate is less precise than ours for several reasons: a) their assumption regarding equal PAL values for all countries (given the very different life-styles prevalent across the world); b) inconsistent weight data across countries coming from different points in time; and c) assuming a constant unchanging BW overtime in making future projections. Assuming equal PAL for all countries likely overestimates food energy requirements for the developed world thereby underestimating their FW, the opposite holds for developing/underdeveloped nations. A comparison of our FW estimates with those of Hic et al. (Table 3) lends support to this intuition. For China and India, Hic et al. provide a waste estimate of 620 and 210Kcal/day/cap

respectively in 2010, while our estimates for 2011 are 329 and 121 Kcal/day/cap respectively. Our approach also implicitly allows the BW in currently poor nations to follow the same trends as those already observed in affluent nations, as they grow richer. Our global estimate for the year 2011 using AIC data from ICP 2011, is 727 Kcal/day/cap. Note that the 727 Kcal/day/cap estimate should be taken as a rough indication, as it (AIC data) should appropriately be adjusted for 2003–2011 USD inflation.

**Hall et al.** [3] **and Buzby et al.** [28]: Two further studies report FW estimates in Kcal terms. Hall et al. study the evolution of food waste in US between 1974–2003 using United States Department of Agriculture (USDA) data on FA, and National Health and Nutrition Examination Survey data on average BW. They show that allocating a fixed fraction of food availability to estimate consumer FW, underestimates FW. As per Hall et al., Americans in 2003 wasted 1400 Kcal/cap/day. In comparison, Buzby et al. estimate the per capita FW in 2010 in US to be 1249 Kcal/day using the same food availability data but assuming a fixed fraction of different commodities being wasted. As suggested by Dou et al. [29], the Buzby et al. estimate of FW is conservative and likely underestimated. Our regression estimates predict 1482 Kcal/day/cap wasted in 2003 in the US, which increased to 1572 Kcal/day/cap by 2011. This indicates, if not establishes, the ability of our approach to get FW estimates similar to ones provided by studies using detailed country specific data. Across all works, Hic et al. estimate the least amount of food wasted in US in 2010, at 1050 Kcal/day/cap (again pointing to possible downward bias in their estimates for developed countries).

**Affluence elasticity results and implications for future evolution of FW.**   The regression estimate [$\partial$FW/$\partial$ln(AIC)$\cong$576.7] of the logarithmic function, along with the world FW estimate of 526 Kcal/day/cap, yields the global affluence elasticity of waste to be 1.09 in 2005. Since the global elasticity is positive and greater than one, consumer FW could be seen as a luxury good–a deduction not entirely against expectations. The country-specific elasticities vary owing to the different initial FW data: ranging from 0.36 (Belgium) to 18.14 (Philippines) in the sample countries. The developing countries with positive FW all have an affluence elasticity above 1, implying that income growth will generate relatively more FW in these countries. The countries with low elasticities ($<$ 1) are ones that already waste a lot of calories.

By the year 2011, as FW increases to 727 Kcal the elasticity falls to 0.79. Fig 4 shows the point estimates of elasticities and FW for sample countries in 2011, ranked by AIC/capita (x-

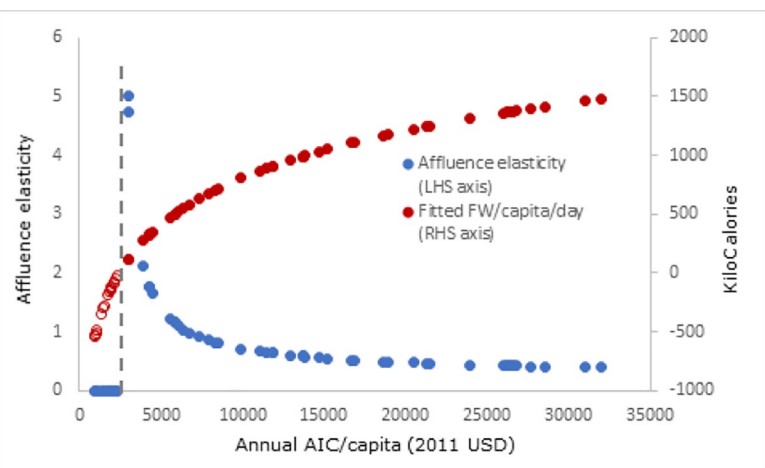

**Fig 4. Food Waste (FW) increases and affluence elasticity declines with increase in affluence.** Source: Food waste and affluence elasticity for sample countries using estimates and ICP data in 2011.

axis). The figure clearly shows FW first rising rapidly and then gradually. At the same time, the affluence elasticity decreases rapidly with rising incomes starting from very high levels. This relationship is similar to the one found between income and calories [30]. The figure also shows a vertical dashed line as the threshold level of AIC/capita beyond which consumer food waste turns positive (shown as change from hollow to solid coloured data points in the per capita food waste series) and the elasticity jumps from non-existent (zero) to very high levels. Even though the affluence elasticities are higher for poor countries, it should not be interpreted as these countries currently wasting a lot of calories. In fact, because they have a low FW to begin with, the responsiveness of waste to increases in expenditure, is high. The lower elasticity can also be an indication of nutrition transition–substituting away from calorie dense foods first to more animal products and finally to fruits and vegetables [31]–and wasting less calorie dense food results in less calorie waste.

Although we have not explicitly accounted for nutrition transition and it is beyond the scope of this work, as higher incomes, health consciousness (following dietary guidelines) and education (knowing dietary guidelines) are highly correlated, the transition to healthier food habits (including eating more fruits and vegetables) is very likely accountable by a mix of these factors [32,33]. And despite not being as well established as Bennet's Law (declining importance of grains in consumption, with rising incomes) [34], the correlation between income and fruits and vegetable intake is touched upon by some scare literature [31,35,36] and needs more work.

Empirically, not accounting for the affluence elasticity of waste leads to overestimates of the consumption response with rising incomes. It has been a problem, for example resulting in anthropometrically impossible projections of food consumption for consumers in countries with strong economic growth [14]. Another example is the estimated calorie intake elasticity for United Kingdom in year 2002 is 0.98 [37], but accounting for waste estimates from the current work and Eq (3), results in a corrected number of 0.81.

## Discussion

Our approach to obtaining quantitative estimates of the link between FW and consumer affluence provides a way of obtaining comparable FW estimates for countries without detailed surveys, based only on easily available consumer expenditure or GDP data. Using this simple approach, we are able to replicate the FW point estimates for US provided by Hall et al. using much less detailed data. Globally we find that consumers were wasting as much as 727 Kcal/day/cap in 2011, rising from 526 Kcal/day/cap in 2005. Just like Hall et al. 2009 show that USDA's consumer FW estimates using fixed waste factors lead to underestimating the numbers for US, our results indicate that FAO's consumer FW estimates of 214 Kcal/day/capita for the year 2005–2007 are grossly underestimating the extent of problem globally. Given that this method could generate globally comparable Food Waste data using readily and publicly available macro and anthropometric data in a transparent manner, it would be very easy to use it for constructing a measurable Food Waste Index to assess current situation regarding the Sustainable Development Goals (SDG 12). More precisely, this method can be used to provide the level of FW that would exist in absence of any intervention measures towards reducing FW in a country. This could provide a scale against which progress towards halving FW as mandated by SDG 12.3 can be assessed, to show how far a nation is from achieving the target.

While we address the fixed factors issue with the FAO approach, we are as limited as FAO in using the FAO FBS data as a measure of food availability. Despite this shortcoming, we find strong evidence of a link between FW and consumer affluence and call it affluence elasticity of waste. Our indicative higher affluence elasticities for growing economies point to a brewing

potential future problem. If these growing economies follow the same growth paths as the developed regions, we will soon see similar FW patterns evolving. According to our estimates, annual per capita consumer expenditure of about 2450 (International 2005 USD) or about $6.70/day/capita, is the level at which policy-makers should start paying particular attention to consumer FW in a country and implement consumer awareness and education programs to counter it before it explodes. The existence of affluence elasticity of food waste so far ignored, has wide implications for both the theory of consumption and its numeric implementation.

Using easily available standard macro-economic data like expenditure, does not undermine the importance of detailed country specific micro-data analysis, to understand the influence of specific factors such as education, consumer awareness, attitudes, cultural food practices and preferences etc.; only that detailed internationally consistent data on such factors is harder to gather.

## Supporting information

**S1 File. Sample dataset and results.**
(XLSX)

**S2 File. Derivation of Eq (1).**
(DOCX)

**S3 File. R readable data and script for linear log model and Fig 3.**
(ZIP)

## Acknowledgments

We would like to thank Professor Tom Hertel (Purdue University) for encouraging us to pursue the idea regarding elasticity of food waste, Dr. Lindsay Shutes and our anonymous reviewers for suggesting changes to this manuscript, and various seminar and conference participants for feedback in the early stages of this work.

## Author Contributions

**Conceptualization:** Monika van den Bos Verma.

**Data curation:** Monika van den Bos Verma, Linda de Vreede.

**Formal analysis:** Monika van den Bos Verma.

**Funding acquisition:** Monika van den Bos Verma, Thom Achterbosch, Martine M. Rutten.

**Investigation:** Monika van den Bos Verma, Linda de Vreede.

**Methodology:** Monika van den Bos Verma.

**Project administration:** Monika van den Bos Verma, Thom Achterbosch, Martine M. Rutten.

**Supervision:** Monika van den Bos Verma, Martine M. Rutten.

**Validation:** Monika van den Bos Verma.

**Visualization:** Monika van den Bos Verma.

**Writing – original draft:** Monika van den Bos Verma.

**Writing – review & editing:** Monika van den Bos Verma, Thom Achterbosch.

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
