## [Decision Letter · Decision Letter 0]

10 Sep 2019

PONE-D-19-21723

Consumers discard a lot more food than widely believed: Estimates of global food waste using consumer affluence and energy gap approach

PLOS ONE

Dear Dr. Verma,

Thank you for submitting your manuscript to PLOS ONE. After careful consideration, we feel that it has merit but does not fully meet PLOS ONE’s publication criteria as it currently stands. Therefore, we invite you to submit a revised version of the manuscript that addresses the points raised during the review process.

We would appreciate receiving your revised manuscript by Oct 25 2019 11:59PM. To enhance the reproducibility of your results, we recommend that if applicable you deposit your laboratory protocols in protocols.io, where a protocol can be assigned its own identifier (DOI) such that it can be cited independently in the future. For instructions see: http://journals.plos.org/plosone/s/submission-guidelines#loc-laboratory-protocols

We look forward to receiving your revised manuscript.

Kind regards,

Taoyuan Wei

Academic Editor

PLOS ONE

2. We note that Figure 1 in your submission contain copyrighted images.

All PLOS content is published under the Creative Commons Attribution License (CC BY 4.0), which means that the manuscript, images, and Supporting Information files will be freely available online, and any third party is permitted to access, download, copy, distribute, and use these materials in any way, even commercially, with proper attribution. For more information, see our copyright guidelines: http://journals.plos.org/plosone/s/licenses-and-copyright.

a.         You may seek permission from the original copyright holder of Figure(s) [#] to publish the content specifically under the CC BY 4.0 license.

Reviewers' comments:

Reviewer's Responses to Questions

**Comments to the Author**

1. Is the manuscript technically sound, and do the data support the conclusions?

Reviewer #1: Partly

Reviewer #2: Partly

Reviewer #3: Yes

2. Has the statistical analysis been performed appropriately and rigorously? 

Reviewer #1: Yes

Reviewer #2: Yes

Reviewer #3: Yes

3. Have the authors made all data underlying the findings in their manuscript fully available?

Reviewer #1: No

Reviewer #2: Yes

Reviewer #3: Yes

4. Is the manuscript presented in an intelligible fashion and written in standard English?

Reviewer #1: Yes

Reviewer #2: Yes

Reviewer #3: No

5. Review Comments to the Author

Reviewer #1: Review for: ‘‘Consumers discard a lot more food than widely believed: Estimates of global food

waste using energy gap approach and affluence elasticity of food waste’’

Summary:

This paper proposes an indirect method for computing internationally comparable consumer food waste. The method computes food waste on food availability, energy gap and consumer affluence. They also establish a relationship consumer affluence and food waste as well as a threshold where this relationship changes. Overall, this is an interesting exercise and piece of contribution to improve our understanding of the quantity of food wasted world-wide.

I have the following major and minor comments:

Major comments:

1. The paper lacks clarity and details on how they are computing some of the estimations. For instance, the authors argue that the relationship between food waste and consumer affluence changes after some threshold. But I was expecting a graph showing this non-linear relationship.

2. The authors claims estimating affluence elasticity of food waste, an idea I liked. But I am not sure if this is executed correctly because quantifying elasticity involves estimating log-log relationships. But the authors are not exactly estimating such relationships.

3. I would add a bit more discussion on the actual estimation of relationships. I was expecting more details on the estimation and tables.

4. Data related issues: I am wondering if the authors can use a combined ICP data for 2005 and 2011, so that they may also estimate panel regressions.

5. Data related issues: why should we trust the FAOSTAT for food availability but not for food waste or food loss? This requires a bit of justification.

Minor comments:

There seems some errors and typos in few places (e.g., page 8)

Reviewer #2: General Comments

This is an interesting study that investigates relation between food waste and consumer affluence. Mainly, the authors estimated food waste as the differences between food availability and food consumption. Food consumption was calculated based on physical activity level and body weight as few other studies also did. Afterwards, the manuscript presents the fitting of the per capita food waste and income data. Although the study presents interesting finding showing that food waste currently increase with income, the manuscript cannot be considered for publication in the current form due to following reason.

First, the method use for estimating food waste has been reported by previous studies which the authors do not acknowledge in the manuscript. Instead, the authors claim that the applied method is a sophisticated one although it does not account for food requirement of children and pregnant and lactating women. This additional food requirement has been considered by previous study. Thus, the manuscript needs to refer to these studies in the beginning wherever it is appropriate.

Second, the claim that this method would be useful to monitor SDG 12 target 12.3 contradicts with the aims of SDGs. SDGs aim to reduce food loss and waste while increasing income. However, this regression provided by this study shows that food waste increase with income.

Third, figure/table and its caption need to be standalone. Currently, this is not the case. Acronyms in figure/table are not explained in the caption.

Fourth, please include an in-text citation, using the author's last name and the year of publication, when the manuscript refers to, summarize, paraphrase, or quote from another source.

Specific Comments

Line 24-25: The studies say that one third of produced food is either lost or wasted. Since this study focuses on food waste, it would be precise to differentiate between food loss and food waste.

Line 25-26: See also the IPCC SRCCL Chapter 5 for food loss and waste in the food system.

Line 27-28: “supply side food waste is constrained by an upper-limit as dictated by food availability” is this “supply side food waste” of food waste by consumers. It would be use single term so that readers are not confused.

Line 34-38: This is unclear what “consumer specific attributes” mean. Global studies such as Hic et al 2016 and Walpole et al BMC Public Health 2012 studies food waste globally based on different country specific attributes. Please acknowledge these previous studies. Additionally, Hic et al 2016 also presents associations between food waste and Human Development Index (HDI).

Line 39-51: It is unclear why results are presented here in introduction instead of research questions or objective. Is this a new scientific writing style? Additionally, Target 12.3 of SDG 12 is about halving the global per capita food waste and reducing food loss. Please make this explicit.

Line 53-54: Better argument for not using FAO data would be lack of country specific values for waste. However, Hic et al 2016 provides country specific values, why the authors does not consider the use them.

Line 70-80: When this study considers constant body weight, the study would be similar to Hic et al 2016 and Walpole et al 2012. Hic et al 2016 also consider food requirement for children and additional calories requirement for pregnant and lactating women. It is unclear how this study advances on the previous studies without considering these factors.

Line 83: It is unclear why the calculated FW data is only of sample countries is. The authors need to justify why they did not attempt to fill the data gap.

Line 86-87: Generally, food availability is associated with affluence or income. However, there are studies which also look at other factors. See Bodirsky et al. 2015 Plos One on Good Food Demand. Please better justify the assumption. This also holds of the relation between food consumption and affluence.

Line 92-93: Need to provide some clarification of what are the sample countries. Please also make it explicit that this is a cross-sectional analysis.

Line 106-109: It is unclear how the above described method can be applied to estimate waste of individual food commodity. How can consumption of individual food commodity be estimated based on energy requirement values?

Line 123-124: It is unclear what does this “upward bias” mean.

Line 131-133: Taking the average is done mainly on crop production to rule out fluctuation in food production. However, food availability is an outcome of food production, trade, stock variation, etc. Need a better justification for this argument.

Line 144-148: Please make it clear that the data is available in supplementary. The supplementary files need to be referred in the main text.

Line 161-164: Taking into account dietary requirement of children is important because demographic characteristic also shapes a country food demand.

Line 166-174: It is unclear why the authors do not attempt to fill the data gap from other sources such as Demographic and Health Survey data. When the data gaps are filled, there might be differences in obtained results.

Line 185: Please refer to the supplementary material in different parts of the method section wherever it is relevant so that reader will obtain additional information from the supplementary.

Line 217-221: This reads like table caption instead of interpretation of the results presented in the table.

Table 2: GDP/Cap has not been mentioned in the method section. Please mention this also there.

Line 231-233: See previous comment on filling data gaps.

Line 233-237: It is unclear how outliers and deviations are handled here. It is not enough to a single value. Figure 2 clearly shows that there are countries above and below the regression lines.

Line 239-241: This has been reported by previous studies, see Walpole et al 2012 and Hic et al 2016.

Line 336: Unclear “nutrition transition” means here.

Line 343-355: Using of this approach for monitoring Target 12.3 of SDG 12 would create more problem than solving it. This is because SDG aims to increase GDP but reduces food loss and waste. Currently, more food is waste with income growth. This is one of the trade-offs in SDGs (see Pradhan et al. 2017 Earth’s Future) which needs to tackle to achieve the 2030 Agenda for Sustainable Development. Thus, it would not be realistic to use the identified regression for 2001-2005 also in the future.

Line 356-364: What about the deviation from the fitting? Authors need to be critical about their findings by clearly providing limitation of the study.

Figure 1: “Physical Activity” instead of “Activity”

Figure 3: Legend missing

Figure 4: Names of Y-axes are missing

Reviewer #3: Review of Verma et al

This paper uses a different measure of food waste at a national scale than has featured in the literature to date. As such, it is a welcome addition to measurement of this important indicator of sustainability. The concept of affluence elasticity of food waste is a useful one in investigating this economic-behavioural relationship.

General Comments

The paper would benefit significantly from an edit that paid careful attention to the use of English language.

In Section 2 and throughout it would be useful to have further discussion of the treatment of uncertainty in the modelling and consequent results. For example, there is a discussion of the bias resulting from exclusion of children in the BW estimates (lines 161-164); how large might this, and other biases, be?

The discussion of the results of affluence is not totally intuitive. I would have thought that elasticities would continue to increase as incomes rise, since there is less of an income constraint on meeting basic calorific needs. But perhaps there is a limit to calorific profligacy, beyond which it falls, giving an inverted-U shape. Either way, it would be worthwhile having more discussion of the phenomenon you observe. Could it be related to the Environmental Kuznets Curve concept? If so, what are the policy implications of your findings? Finally, has the transition to fruit and vegetables been observed anywhere? If so, it would be good to know where, and to what it can be attributed.

Are the non-linear elasticities used in the global projection estimates. It would be helpful to make this clear.

Specific comments

Line 16. Is the per day/capita level of expenditure just on food or on all consumer goods & services?

Line 73. In relation to equation 1 would it be worth stating explicitly that – as I understand it - its validity depends on the characteristics of the population remaining the same over time?

Line 124. Is it possible to say anything about the broad size – in percentage terms - of the upward bias?

Page 8. The formatting is lost here. Also, the text seems to be incomplete (e.g. line 223)

Lines 235-237. Is the increase in Kcal/day/capita between 2005 and 2011 as a result of increases in incomes? Or is there another explanation?

Lines 265-267. The argument is not quite clear to me. Are you referring to a potential double-counting estimation procedure?

Line 308. Maybe say explicitly why the Buzby et al estimate of FW is conservative?

Line 356. Does the use of the FA data from FAO lead to over-estimates? Either way, it would be good to make the direction of the bias clear.

Lines 363-368. It would be good to have a little more on the advantages of using this new measure, in terms of its policy applications.

6. PLOS authors have the option to publish the peer review history of their article (what does this mean?). If published, this will include your full peer review and any attached files.

Reviewer #1: No

Reviewer #2: No

Reviewer #3: No

---

## [Author Response · Author response to Decision Letter 0]

17 Oct 2019

We have responded to all the comments made by the reviewers and addressed the journal requirements pointed out to us. The responses are submitted as a separate pdf file.

---

## [Decision Letter · Decision Letter 1]

22 Nov 2019

PONE-D-19-21723R1

Consumers discard a lot more food than widely believed: Estimates of global food waste using energy gap approach and affluence elasticity of food waste

PLOS ONE

Dear Dr. Verma,

Thank you for submitting your manuscript to PLOS ONE. After careful consideration, we feel that it has merit but does not fully meet PLOS ONE’s publication criteria as it currently stands. Therefore, we invite you to submit a revised version of the manuscript that addresses the points raised during the review process.

As you can see below, there are still some major concerns from the reviewers on your revised manuscript. Hence, I would like you to revise again by considering their comments. Please also notice that the journal does not have any requirements for or against using bullet points in the Introduction of a manuscript to present the summary of conclusions.

We would appreciate receiving your revised manuscript by Jan 06 2020 11:59PM. To enhance the reproducibility of your results, we recommend that if applicable you deposit your laboratory protocols in protocols.io, where a protocol can be assigned its own identifier (DOI) such that it can be cited independently in the future. For instructions see: http://journals.plos.org/plosone/s/submission-guidelines#loc-laboratory-protocols

We look forward to receiving your revised manuscript.

Kind regards,

Taoyuan Wei

Academic Editor

PLOS ONE

Reviewers' comments:

Reviewer's Responses to Questions

**Comments to the Author**

1. If the authors have adequately addressed your comments raised in a previous round of review and you feel that this manuscript is now acceptable for publication, you may indicate that here to bypass the “Comments to the Author” section, enter your conflict of interest statement in the “Confidential to Editor” section, and submit your "Accept" recommendation.

Reviewer #1: All comments have been addressed

Reviewer #2: (No Response)

Reviewer #3: (No Response)

2. Is the manuscript technically sound, and do the data support the conclusions?

Reviewer #1: Yes

Reviewer #2: Partly

Reviewer #3: Yes

3. Has the statistical analysis been performed appropriately and rigorously? 

Reviewer #1: Yes

Reviewer #2: Yes

Reviewer #3: Yes

4. Have the authors made all data underlying the findings in their manuscript fully available?

Reviewer #1: No

Reviewer #2: Yes

Reviewer #3: Yes

5. Is the manuscript presented in an intelligible fashion and written in standard English?

Reviewer #1: Yes

Reviewer #2: Yes

Reviewer #3: No

6. Review Comments to the Author

Reviewer #1: The authors have mostly addressed my major concerns and comments. I believe this piece can contribute to the scant but evolving literature on food wast and food loss measurement.

Reviewer #2: General Comments

I thank the authors for their attempt to the address the comments. Although the authors provided responses to all the comments, they did not incorporate the changes in the revised manuscript. This is also visible while looking at the track changed version of the revised manuscript. Thus, I suggest considering the manuscript for publication after addressing and incorporating the following comments in the revision process.

First, as mentioned in the previous round, the method use for estimating food waste has been reported by previous studies which the authors do not properly acknowledge in the manuscript. Instead, the authors claim that this has not been done yet. Line 53-56 says:

“The demand side requires data on consumer attributes such as income, age, gender, education, residence etc. While there are individualistic attempts to capture the impact of consumer specific attributes [8,9] there is no study attempting this at a global level. This work is a first attempt in linking the amount of food wasted to one such consumer attribute.”

This is wrong because global studies such as Hic et al 2016 and Walpole et al 2012 cover the aspect of age, gender, and body weight. The authors need to acknowledge this.

Second, the study is based on the sample countries. However, the manuscript does not tell about these countries in the very beginning. As a reader, I would like to know which these sample countries are in the first place it is mentioned (see Line 99). Since readers do not know about the features of these sample countries, it is hard to justify the argument present in Line 113-114.

Third, I also request the authors to be critical in their projections. For example, in 2011 the global food availability is 2869 kcal/cap/day and the estimated of food waste by this study is 727 kcal/cap/day. This means 2142 kcal/cap/day is the global food consumption. This global value is below the estimated dietary requirements.

Fourth, the recent report from FAO, The State of Food and Agriculture (SOFA), focuses on food loss and waste. Thus, it would be relevant to discuss the finding of this study also in the context and finding from SOFA.

Specific Comments

Line 115: What does “Genetic differences” mean? Please explain this to reader.

Line 232-233: Since the study used the average value between 2000 and 2005, it would be better to say “circa (ca.) 2003 instead of 2003.

Line 247: GDP/Cap has not been mentioned in the method section. Please mention this also there. The authors’ repose was “But perhaps this intention of using it as sensitivity test, needs to be made more clear, and this has now been done”. However, when this is used as sensitivity test, this needs to mention in method. There is no sentence on “sensitivity test” in the method section.

Line 269-270: When possible it would be better to provide range or standard deviation for the estimates

Line 272-274: This is no surprise and the study use the relation of increase food waste with increase income. However, the relation is not able to capture rich countries wasting less food, e.g., Japan.

Line 374: Although in the response, the authors mentioned: “It is defined a few lines above as (quoting from text) – “nutrition transition– substituting away from calorie dense foods first to more animal products and finally to fruits and vegetables”.”. This change is not visible in the manuscript.

Line 383-385: Please highlight that this is the case under the current trend.

Line 392-394: Regarding food waste index, please also see SOFA 2019.

Line 395-397: The authors may include that “SDG 12 (Responsible Consumption and Production) could be the bottleneck for sustainable development (Pradhan et al. 2017)”

Reviewer #3: Thank you for your responses to my comments. They are reassuring. Your response to my query about the transition to fruit and vegetables is useful; it would be good to include a condensed version of this in the text. I also think a revision by a native English speaker is still required.

7. PLOS authors have the option to publish the peer review history of their article (what does this mean?). If published, this will include your full peer review and any attached files.

Reviewer #1: No

Reviewer #2: No

Reviewer #3: No

---

## [Author Response · Author response to Decision Letter 1]

7 Jan 2020

Responses are provided in a separate file: 'Response to Reviewers.docx'

---

## [Editor Report · Decision Letter 2]

15 Jan 2020

Consumers discard a lot more food than widely believed: Estimates of global food waste using an energy gap approach and affluence elasticity of food waste

PONE-D-19-21723R2

Dear Dr. Verma,

We are pleased to inform you that your manuscript has been judged scientifically suitable for publication and will be formally accepted for publication once it complies with all outstanding technical requirements.

With kind regards,

Taoyuan Wei

Academic Editor

PLOS ONE

---

## [Editor Report · Acceptance letter]

17 Jan 2020

PONE-D-19-21723R2 

Consumers discard a lot more food than widely believed: Estimates of global food waste using an energy gap approach and affluence elasticity of food waste 

Dear Dr. Verma:

I am pleased to inform you that your manuscript has been deemed suitable for publication in PLOS ONE. Congratulations! Your manuscript is now with our production department. 

With kind regards,

on behalf of

Dr. Taoyuan Wei 

Academic Editor

PLOS ONE